# GRAPH NEURAL NETWORKS FOR AERODYNAMIC FLOW RECONSTRUCTION FROM SPARSE SENSING

## ABSTRACT

Sensing the fluid flow around an arbitrary geometry entails extrapolating from the physical quantities perceived at its surface in order to reconstruct the features of the surrounding fluid. This is a challenging inverse problem, yet one that if solved could have a significant impact on many engineering applications. The exploitation of such an inverse logic has gained interest in recent years with the advent of widely available cheap but capable MEMS-based sensors. When combined with novel data-driven methods, these sensors may allow for flow reconstruction around immersed structures, benefiting applications such as unmanned airborne/underwater vehicle path planning or control and structural health monitoring of wind turbine blades. In this work, we train deep reversible Graph Neural Networks (GNNs) to perform flow sensing (flow reconstruction) around two-dimensional aerodynamic shapes: airfoils. Motivated by recent work, which has shown that GNNs can be powerful alternatives to mesh-based forward physics simulators, we implement a Message-Passing Neural Network to simultaneously reconstruct both the pressure and velocity fields surrounding simulated airfoils based on their surface pressure distributions, whilst additionally gathering useful farfield properties in the form of context vectors. We generate a unique dataset of Computational Fluid Dynamics simulations by simulating random, yet meaningful combinations of input boundary conditions and airfoil shapes. We show that despite the challenges associated with reconstructing the flow around arbitrary airfoil geometries in high Reynolds turbulent inflow conditions, our framework is able to generalize well to unseen cases.

## 1 INTRODUCTION

Many engineering applications stand to benefit from the ability to sense and reconstruct fluid flow features from sparse measurements originating at a structure's surface. Flow sensing could be crucial for improvements in the accuracy and resilience of wind turbine and unmanned aircraft controllers. Another possible application is monitoring of wind loaded structures (Barber et al., 2022), where the use of cheap micro-electromechanical systems (MEMS) in combination with novel methods for flow sensing could lead to robust structural health monitoring solutions. In this work, we focus on common aerodynamic structures: we aim to reconstruct the flow around 2-D airfoils. Traditionally, computing the flow around an airfoil requires approaches from Computational Fluid Dynamics (CFD), which are forward-physics simulators. In CFD, the inflow, outflow and wall boundary conditions are set, and over many iterations a solution for the discretized Navier-Stokes PDEs is reached, which then yields a pressure distribution at the airfoil surface. We aim to solve the inverse problem: given only the pressure distribution at the airfoil surface, a solution for the flow field and farfield boundary conditions is to be found. Moreover, our aim is to do so for any airfoil geometry subject to a wide variety of turbulent inflows.

Adopting the notation of Erichson et al. (2020), the problem can be described in the following manner. An airfoil equipped with $p$ distributed barometric sensors is placed in a steady flow of air, providing surface pressure measurements $s \in \mathbb{R}^p$ at multiple locations around its perimeter. The sensors sample from the surrounding flow field $x \in \mathbb{R}^m$ through a measurement operator $H$:

$$s = H(x) \tag{1}$$

The goal is to construct an estimate of the flow field $\hat{x}$ surrounding the airfoil, by learning from training data a function $\mathcal{F}$ that approximates the highly nonlinear inverse measurement operator $G$ such that:

$$\mathcal{F}(s) = \hat{x} \approx x = G(s) \tag{2}$$

Meshes are an extremely useful tool, indispensable in many engineering domains and especially in CFD. Contrary to Cartesian grid representations, mesh representations offer high flexibility for irregular geometries and allow for variable spatial density. This makes them ideal for discretizing complex physical problems, where one can balance the trade-off between numerical accuracy and computational efficiency in certain regions of interest. Furthermore, meshes can also be described in terms of nodes and edges, i.e. as a graph. In this context, the flow reconstruction problem can be described as follows. A graph $\mathcal{G} = (\mathcal{V}, \mathcal{E}, \mathcal{U})$ is constructed from an airfoil CFD mesh, with $m$ fluid nodes $\mathcal{V}_f$ and $p$ airfoil boundary nodes $\mathcal{V}_a$. The flow features of all $\mathcal{V}_f$ are unknown, whilst the features of $\mathcal{V}_a$ are known. Our aim is then to learn a graph operator $\mathscr{F}$ that estimates the information at the fluid nodes using the information contained at the airfoil boundary nodes, the input graph-level attributes $\mathcal{U}_{in}$ and the edges $\mathcal{E}$:

$$\hat{\mathcal{V}}_f = \mathscr{F}(\mathcal{V}_a, \mathcal{E}, \mathcal{U}_{in}) \tag{3}$$

An ancillary goal is to estimate the global context $\mathcal{U}$ of the graph, as this contains relevant information for applications. Figure 1 provides an description of the flow reconstruction problem in terms of graph learning.

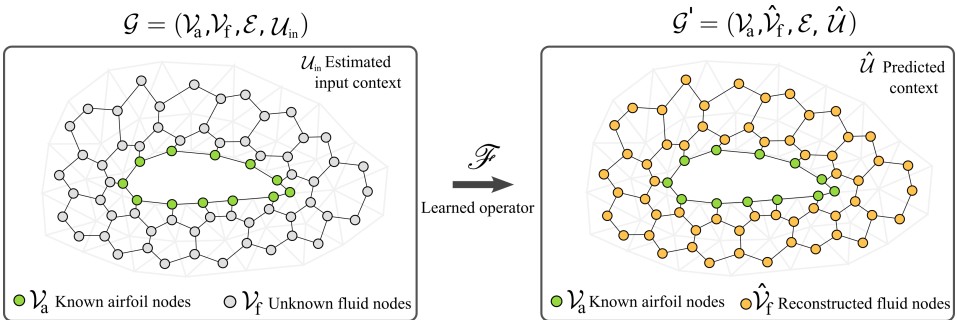

Figure 1: Problem setup. We aim to learn the reconstruction operator $\mathscr{F}$ which estimates properties of the fluid nodes as well as the graph context. This amounts to reconstructing a solution to the Navier-Stokes equations which satisfies the boundary conditions perceived at the airfoil surface.

From a geometric learning perspective, flow reconstruction is a challenging problem for several reasons. The first significant hurdle to overcome is the size of our graphs. We use meshes with high densities close to the airfoil in order to achieve good spatial resolution in these critical regions. Thus, our dataset contains graphs with a mean of around 55'000 nodes, which is an order of magnitude higher than previous mesh-graph learned simulation methods (Pfaff et al., 2020). Moreover, the input information is concentrated in a very localized domain of the graph: the airfoil nodes. It is difficult to propagate the necessary information to reconstruct nodes far from the airfoil with a shallow Graph Neural Network (GNN), meaning that deep GNN architectures with a large number of message-passing steps are required to push this 'information barrier' away from the airfoil nodes. However, deep GNNs go hand in hand with other issues such as large memory requirements, over-smoothing, and over-squashing.

In this work, we combine a number of existing graph-learning methods to tackle the aforementioned challenges. Our contributions may be summarized as follows:

- We combine unknown Feature Propagation (Rossi et al., 2021) with very deep Grouped Reversible GNNs (Li et al., 2021) to reconstruct flow features at the fluid nodes, whilst additionally gathering contextual farfield information.

- Generalization of 2D aerodynamic flow field learning with GNNs, including (1) arbitrary airfoil geometries, (2) arbitrary turbulent inflow conditions (flow velocity, turbulence intensity, and angle of attack), and (3) simultaneous flow field reconstruction and inference of contextual farfield flow information based only on sparse pressure data on the surface of the airfoil.

- Generation of a unique training dataset of OpenFOAM airfoil CFD simulations with many different geometries and inflow conditions which are parsed to graph structure and made publicly available.

- We gather qualitative and quantitative results on unseen airfoil and flow configurations, and perform a number of experiments to understand the limitations of this framework. In particular, we test and compare three different GNN layer architectures.

## 2 RELATED WORK

Machine learning methods have recently garnered interest in the fluid mechanics community (Brunton et al., 2020). Fluid-related problems are typically nonlinear, complex and generate large amounts of data, all of which are conditions under which deep learning approaches thrive. Specifically in the context of flow reconstruction from sparse measurements, several neural network approaches can be found in the literature. In an article by Erichson et al. (2020), a "Shallow Neural Network", i.e. a fully-connected network with only two hidden layers, was applied to estimate transient flows from sparse measurements. The authors trained the networks on a single specific geometrical flow configuration, for example the flow behind a cylinder, and then tested on the same configuration at different time steps. We aim to avoid this limitation as our goal is to estimate the flow around any airfoil geometry. The authors compare their findings against typically used proper orthogonal decomposition (POD) methods and note significant improvements in terms of the reconstruction error. This work was then extended to turbulent flow reconstruction around airfoils based on experimental data in Carter et al. (2021), where the results were compared to Particle Image Velocimetry (PIV) measurements. The viability of neural networks over other approaches was further confirmed in the work of Fukami et al. (2020), where multiple methods were pitted against each other to estimate the flow behind a cylinder and an airfoil. Again, here the models were trained and tested on a single flow configuration, while also being dependent on Cartesian geometrical inputs. In Özbay & Laizet (2022), researchers attempt to avoid this limitation by utilizing Schwarz–Christoffel mappings to sample the points at which the flow is reconstructed, thus rendering the method geometry invariant. The authors train multiple neural network architectures on a collection of transient flow simulations around randomly generated 2-D Bezier shapes at a predefined inflow Reynolds number. As inputs for the flow reconstruction, they use multiple pressure sensors on the shapes' surface as well as velocity probes in the wake. Their results indicate that, when compared to a Cartesian sampling strategy, a significant performance boost is achieved for all neural network types, especially in the vicinity of the immersed shape. While this work demonstrates robustness to various geometric configurations, it requires additional velocity sensors and is trained on a singular farfield boundary condition, both of which we aim to avoid and improve upon. Another method which avoids geometrical dependency is reported in Chen et al. (2021). In this work, to which our approach most closely relates to, the authors utilize a Graph Convolutional Network (GCN) (Kipf & Welling, 2016) on graphs constructed from CFD meshes of randomly generated Bezier shapes. The GCN is used to predict the flow around the shapes at a fixed laminar (Reynolds number of 10) inflow condition without using surface measurements. In our approach, we aim to reconstruct a wide variety of turbulent flows given only surface readings, a significantly less constrained problem. We also aim to characterize the global properties of the flow, similarly to Zhou et al. (2021). To our best knowledge, we are the first to attempt to simultaneously reconstruct the flow while estimating turbulent inflow parameters at large Reynolds numbers for arbitrary airfoil geometries.

The dataset that we generate to train our GNN model is similar in terms of the geometries, meshing and CFD pipeline to the work of Thuerey et al. (2020), the main differences being the chosen RANS model and the post-processing (graph parsing). Other datasets found in the literature focus only on the NACA family of airfoils (Schillaci et al., 2021).

Graph networks are based on the message-passing framework (Gilmer et al., 2017), where a nodes features are updated by aggregating messages emanating from its neighbors. Many different types of message-passing schemes can be constructed, with some using attention mechanisms (Veličković et al., 2017) and others relying on strong theoretical backgrounds (Xu et al., 2018). Graph learning methods are increasingly being applied to a wide variety of physics problems (Sanchez-Gonzalez et al., 2018; 2020). In Pfaff et al. (2020), the authors successfully demonstrate how GNNs can learn to replicate forward mesh-based physics simulators and are able to predict the evolution of a transient solution. Motivated by these results, our approach is constructed upon the the same basic Encode-Process-Decode network structure. However, a key difference to note is that, contrary to the next-step

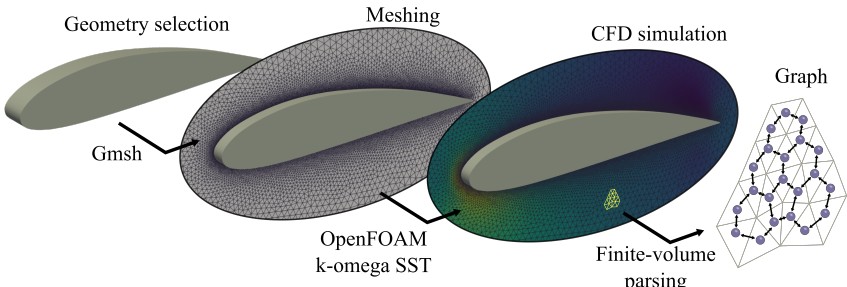

Figure 2: Illustration of the dataset generation pipeline. Airfoil shapes are selected at random from a database, then meshed and simulated in OpenFOAM with random feasible boundary conditions. In the last step, the finite-volume scheme is used to parse the simulation mesh into a graph.

prediction problem, flow reconstruction has to overcome high amounts of missing information, with known features being extremely localized. To address this hurdle, we turn to graph-based feature propagation methods (Rossi et al., 2021), which is closely related to matrix completion approaches (Monti et al., 2017). Feature propagation is an effective yet computationally inexpensive method for initializing graphs with missing features. We use this method as pre-processing step, through which graphs are passed before being fed into the rest of the GNN model.

Training GNNs for very large graphs is challenging, with typical approaches tending toward minimizing the number of learnable parameters so that the problem becomes tractable (Chen et al., 2020). This often results in relatively shallow GNNs, which could adversely influence the propagation of the information contained at the airfoil nodes through sufficient extents of the graph. Making use of subgraph sampling strategies (Hamilton et al., 2017) is another possible approach, one which also allows for larger/deeper GNNs. However, these methods are not applicable in our case, as we need to feed entire graphs in one pass due to the heterogeneity in information localization. Moreover, subgraph sampling would yield additional difficulties for our ancillary goal of predicting global graph properties for farfield estimation. Recent work by Li et al. (2021) has shown that it is possible to train very deep GNNs on large graphs by making use of Grouped Reversible layers, which reduces memory requirements at the cost of extra computation. This method forms the core of the processing block of the proposed flow reconstruction GNN. Another issue which traditionally characterizes training deep GNNs on large graphs is over-squashing (Alon & Yahav, 2020). Over-squashing is a by-product of a graph's structure, where bottlenecks and tree-like structures (Topping et al., 2021) can cause the latent representation of certain nodes to be overwhelmed by the amount of information needed to be stored. We make use of this information while parsing the simulation meshes into graphs.

## 3 DATASET GENERATION

In this section we introduce the different elements of our data generation pipeline. In total we generate 1120 converged simulations, which are separated into train, validation and test datasets in a 80/10/10 split. Figure 2 depicts an illustration of this pipeline.

**Geometry selection and meshing** In our dataset generation pipeline, airfoil shapes are drawn at random from the UUIC database of airfoils (Selig, 1996). Before passing the shape to the meshing algorithm, we carry out some additional interpolation and processing to make sure that the selected airfoil has a sufficient amount of points at the leading-edge as well as a properly defined trailing-edge. Then, we use Gmsh (Geuzaine & Remacle, 2020) to construct an unstructured O-grid type mesh around the selected airfoil. A sizing field is set close to the airfoil in order to make sure that meshes with appropriate y+ values for the CFD wall-functions are generated. An overall sizing parameter is also set for sufficient farfield density, but is adjusted to ensure that an acceptable amount of cells are created ($< 150'000$).

**CFD simulations** Each mesh is associated with a different inflow configuration. Three parameters control the farfield conditions: angle of attack, inflow velocity and turbulence intensity. These parameters are drawn from probability distributions reflecting realistic atmospheric flows at Reynolds

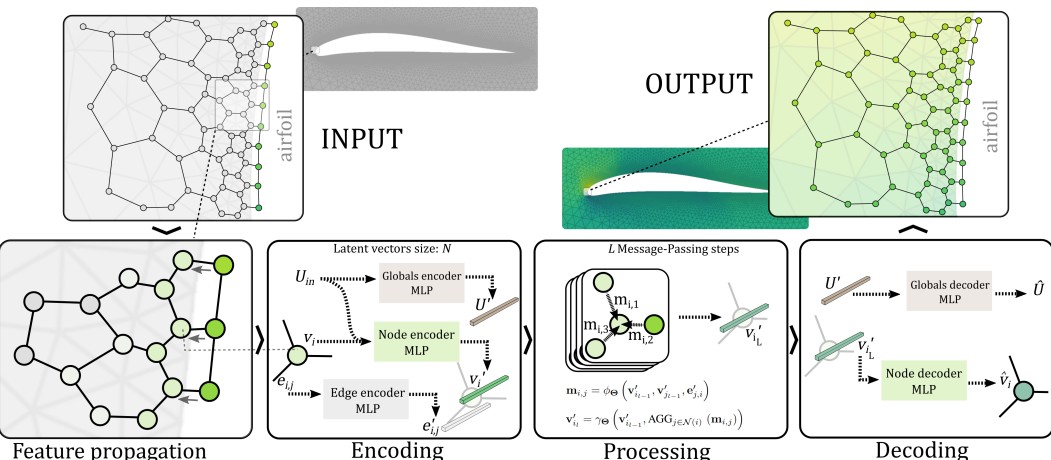

Figure 3: Overview of the Flow Reconstruction GNN achitecture. Feature Propagation is used to initialize the unknown fluid nodes. The graph is then passed through the Encode-Process-Decode pipeline, to obtain the reconstructed flow. During the Process step, node features are updated via message-passing within a deep Grouped Reversible GNN.

numbers ranging from $2 \cdot 10^5$ to $6.5 \cdot 10^6$, with a mean Reynolds number of around $3 \cdot 10^6$. This mean value is well beyond the typical laminar-to-turbulent transition threshold of around $5 \cdot 10^5$ (Incropera et al., 1996), which greatly increases the difficulty of the flow reconstruction problem. The farfield conditions form the global context vectors of our graphs and are estimated at inference time. We simulate the flow around the airfoils using a steady 2-D Reynolds-Averaged Navier–Stokes (RANS) CFD solver with the OpenFOAM software package (Jasak et al., 2007). For turbulence modelling, we select the K-Omega SST model (Menter & Esch, 2001) along with the standard OpenFOAM wall-functions for boundary layer treatment. Only sufficiently converged CFD simulations with pressure, velocity and turbulence residuals below $5 \cdot 10^{-5}$ are kept.

**Graph parsing**   Contrary to many other mesh-based physics simulators, CFD solvers such as OpenFOAM are based on finite volume methods. It is a significant difference that should be reflected in the manner in which a mesh is converted into a graph. To do so, we use the cells themselves as the nodes, with bidirectional edges being formed between adjacent cells. This allows us to gather an additional edge feature which is relevant to the underlying physics. Specifically, the length (or surface for 3-D meshes) of the boundary between two cells is used as an edge feature. The benefit of this is twofold: a form of sizing is fed to the network and a quantity relevant to flux computation is set on the edges. For the nodes, we gather 4 types of features: pressure, x-velocity component, y-velocity component and node category (fluid, farfield, wall), the latter of which is inputted as a one-hot vector. The global context features are the farfield conditions (turbulence intensity, inflow velocity and angle of attack). To avoid unnecessary computational overhead, we do not parse the entire CFD domain, which has a radius of 100 airfoil chord lengths, into a graph. Instead we opt to only keep cells within a 1 chord circle centered on the airfoil. Furthermore, we set the airfoil nodes to be located at the meshed airfoil boundary and add bidirectional edges between adjacent airfoil nodes. These additional edges are created with the aim of avoiding tree-like structures in our graphs, as these could potentially cause bottlenecks for the learning process (Topping et al., 2021). The graphs of our dataset have on average around 55k nodes and around 85k individual edges.

## 4 GRAPH NEURAL NETWORK FRAMEWORK

### 4.1 ARCHITECTURE

For our GNN architecture, we adopt the Encode-Process-Decode logic that is now popular for learning on graph-based physics problems (Sanchez-Gonzalez et al., 2020; Pfaff et al., 2020; Godwin et al., 2022), albeit with some notable modifications. Figure 3 shows an overview of the Flow Reconstruction GNN.

**Input features**  In the flow reconstruction problem, we assume that the global context of the graph is unknown, however some useful physical parameters describing the flow can be estimated. Using Bernoulli's principle, and given that all airfoils are simulated with a zero farfield static pressure, the farfield inflow velocity magnitude can be initially approximated as:

$$\hat{U}_\infty = \sqrt{\frac{2 \cdot p_0}{\rho}} \tag{4}$$

where $\rho$ is the density of air (constant throughout simulations) and $p_0$ is the total pressure measured at the stagnation point, which can be estimated by taking the maximum pressure at the airfoil nodes $p_0 = \max(p_{\mathcal{V}_a})$. While Bernoulli's principle is not valid for turbulent flows such as the ones we try to reconstruct, it serves as a good starting point for farfield velocity estimation. Another useful graph property that can be extracted is the normal force coefficient acting on the airfoil. While the lift coefficient is usually used to characterize airfoils, it cannot be calculated as the angle of attack is unknown (a quantity to be inferred from the learned model). Nevertheless, the normal coefficient is directly related to the lift coefficient and brings additional physical information which may aid the network to reconstruct the flow. It can be estimated via the following equation:

$$\hat{C}_n = \frac{\sum\limits_{v_i \in \mathcal{V}_a} p_{v_i} \cdot l_{v_i} \cdot n_{y,v_i}}{p_0} \tag{5}$$

where $l$ is the boundary length and $n_y$ is the y component of the normal boundary vector, both of which are known properties for each mesh cell boundary. We therefore use $\mathcal{U}_{in} = (\hat{U}_\infty, \hat{C}_n)$ as the two-dimensional input context vector.

For the nodes, we only have access to the pressure distribution at the surface of the airfoil, while it is set to 'NaN' values at the fluid nodes. The type of each node is known and is encoded as a one-hot vector, bringing the total number of input node features to four. To account for mesh geometry, the following four edge features are used as inputs: the x and y components of the relative edge direction vector, the edge length, and the cell boundary length value $l$ (see Section 3).

**Pre-processing**  Both the input and target node features are normalized. To avoid biasing the normalization, all pressure features are normalized by the mean and standard deviation of the known airfoil surface pressure distribution, while both components of the velocity target features are normalized by the initial estimated farfield velocity $\hat{U}_\infty$. We use Feature Propagation (Rossi et al., 2021) as a preliminary step before feeding a graph to our GNN. This step is an important part of our framework as it conditions the input graph into a plausible initial state. Essentially, the feature propagator radiates surface pressure information outwards. In most cases, we found 20 feature propagation iterations to be sufficient.

**Encoding**  In the encoding layer, ReLU activated MLPs with two hidden layers and LayerNorm are used to project the input features of the graph into latent vectors of size $N$. This encoding layer differs to the standard GraphNet encoder (Sanchez-Gonzalez et al., 2020) in that the node encoder MLP takes as input the input node features as well as the latent global vector. We make this modification so that graph-level attributes are taken into account in the construction of the node latent variables, as this is not the case in the processing steps.

**Processing**  For the processing step, we opt to use a deep Grouped Reversible GNN (Li et al., 2021) with $L$ message-passing layers. This architecture makes modifications to the typical GNN architecture by first splitting the input node feature matrix $V$ across the feature dimension into $C$ groups $\langle V_1, V_2, ..., V_C \rangle$, which are then processed into grouped outputs $\langle V'_1, V'_2, ..., V'_C \rangle$ with a Grouped Reversible GNN layer. These outputs are computed as follows:

$$\begin{aligned} V'_0 &= \sum_{k=2}^{C} V_k \\ V'_k &= f_{wk}(V'_{k-1}, A, E) + V_k, \quad k \in \{1, \dots, C\} \end{aligned} \tag{6}$$

with $A$ the adjacency matrix and $E$ the edge feature matrix.

The Grouped Reversible framework allows for any type of message-passing architecture to be chosen

for the GNN layer $f_{wk}$. We choose to test three popular types of GNN layers: the Graph Attention Network (GAT) (Veličković et al., 2017), the modified Graph Isomorphic Network (GIN) (Xu et al., 2018) which accounts for edge features (Hu et al., 2019), and the Generalized Aggregation Networks (GEN) Li et al. (2020) which modifies the standard GCN with different aggregation schemes while also utilizing edge features.

**Decoding** Only the nodes and global context are decoded back into feature space, as the edges are not updated. Both decoding neural networks are MLPs with two hidden layers and ReLU activations without any output normalization. At the output of the decoder, we gather for each node the estimated pressure and velocity fields. The output context vector is composed of an updated version of the farfield velocity, as well as an estimation of the inflow angle (angle of attack) and of the turbulence intensity.

## 4.2 TRAINING

**General aspects** Our models are trained on a dataset composed of 896 graphs. Models are trained with the Adam optimizer on a single Nvidia GPU with 10GB of VRAM. Due to the size and nature of the graphs, we can only use a batch size of one, albeit with random order shuffling occurring at each epoch. The learning rate is initially set at $5 \cdot 10^{-4}$ and is exponentially decayed by a factor of $0.97$.

**Loss function** We use a multi-component loss function, in order to minimize both the node feature reconstruction error and the context vector prediction error, with $L_2$ losses for both components. An additional loss component based on the velocity divergence was also tested but yielded too many artefacts and was therefore discarded. The overall loss is:

$$\mathcal{L} = L_2(\mathcal{V}, \hat{\mathcal{V}}) + \lambda \cdot L_2(\mathcal{U}, \hat{\mathcal{U}}) \tag{7}$$

where $\lambda$ is a hyperparameter used to balance the different components. In practice, we usually set $\lambda$ to 1.

## 5 RESULTS

Our trained models are tested on a dataset comprising of 112 unseen airfoil simulations, each with a different combination of turbulent inflow parameters. We show here qualitative and quantitative results for our models and perform experiments aiming to investigate limitations and improvements of the proposed framework.

**Comparison of GNN layers** In Table 1, we gather and compare results for the three different types of GNN layers used in the Processor: GAT, GIN and GEN. The architecture of the Encoder and Decoder networks were kept constant, with a latent size for the node, edge and global features of $N = 128$. In the Grouped Reversible Processor, the number of Layers was set to $L = 30$, while the number of groups was chosen as $C = 4$. To obtain a consistent number of learnable parameters, the hyperparameters for each GNN layer type were carefully selected, more information about the different configurations can be found in the appendix. We also study the impact of the depth and width on the performance of each model in Appendix D.

**Velocity reconstruction** One of the more challenging prediction tasks is the estimation of the velocity field away from the airfoil. Accurately capturing velocity shear and recirculation regions is non-trivial even for CFD simulators and is highly dependant on the airfoil shape and the inflow angle. To complete this task sucessfully, the GNN needs to be expressive enough to propagate relevant information throughout the graph. Table 2 summarizes the prediction errors of the velocity field in multiple concentric regions around the airfoil for the three models. We observe that for all three models, the error on the x-velocity decreases further away from the airfoil, but this is not the case for the y-velocity.

**Qualitative results** Figure 4 displays some qualitative results for our two best performing models (revGAT and revGIN), compared to the CFD simulation ground truth. These results highlight the fact

Table 1: Root mean squared prediction errors averaged over the test dataset for both the flow reconstruction task and the global parameter estimation task. Results are averaged over 3 runs with different initializations. All models have latent size of $N = 128$ and $L = 30$ layers. While the revGAT model performs the best in terms of pressure and y-velocity reconstruction, it is outperformed by the revGIN model when it comes to x-velocity reconstruction, and graph-level attribute prediction.

| | Node reconstruction RMSE | | | Global parameter prediction RMSE | | |
| | pressure | x-velocity | y-velocity | farfield velocity | angle of attack | turbulence intensity |
| | [Pa] | [m/s] | [m/s] | [m/s] | [°] | [-] |
| revGAT | **77.98 $\pm$ 17.99** | 7.96 $\pm$ 1.19 | **2.69 $\pm$ 0.53** | 0.92 $\pm$ 0.17 | 4.16 $\pm$ 0.10 | 0.04 $\pm$ 0.003 |
| revGIN | 158.37 $\pm$ 5.16 | **6.23 $\pm$ 0.32** | 4.43 $\pm$ 0.22 | **0.45 $\pm$ 0.06** | 4.12 $\pm$ 0.09 | **0.03 $\pm$ 0.003** |
| revGEN | 137.51 $\pm$ 17.31 | 10.81 $\pm$ 0.25 | 5.47 $\pm$ 0.05 | 0.46 $\pm$ 0.05 | **4.12 $\pm$ 0.01** | 0.04 $\pm$ 0.002 |

Table 2: Root mean squared prediction errors of the velocity components for different concentric regions of the flow, averaged over the test dataset. Each region is defined as the interior of an ellipse with $a$ the length of semi-major axis and $b$ the length of semi-minor (in chord lengths). Results are averaged over 3 runs with different initializations.

| | | x-velocity [m/s] | | | y-velocity [m/s] | |
| | revGAT | revGIN | revGEN | revGAT | revGIN | revGEN |
| region 1 (a=0.6, b=0.1) | 8.26 $\pm$ 1.34 | **6.46 $\pm$ 0.39** | 11.44 $\pm$ 0.25 | **2.48 $\pm$ 0.60** | 4.15 $\pm$ 0.18 | 5.40 $\pm$ 0.03 |
| region 2 (a=0.7, b=0.15) | 7.98 $\pm$ 1.25 | **6.30 $\pm$ 0.35** | 11.00 $\pm$ 0.26 | **2.49 $\pm$ 0.43** | 4.40 $\pm$ 0.23 | 5.52 $\pm$ 0.05 |
| region 3 (a=0.8, b=0.2) | 7.95 $\pm$ 1.24 | **6.27 $\pm$ 0.33** | 10.92 $\pm$ 0.26 | **2.61 $\pm$ 0.55** | 4.44 $\pm$ 0.23 | 5.53 $\pm$ 0.05 |

that the learned model is able to reconstruct flow features well, albeit with some artefacts. As the distance to the airfoil increases, these defects become more apparent. Moreover, some parts of the flow are not well captured. This is the case for flows exhibiting long wakes. On the other hand, we notice that flow features near the leading edge of the airfoil are in general well captured. We provide additional examples of reconstructed flows in Appendix E.

**Farfield estimation** Figure 5 displays the graph-level context prediction results evaluated on the test set for the revGIN model. The GNN is able to accurately predict farfield inflow velocity, owing to the good initial farfield estimation provided as an input. For the angle of attack estimation, we observe good results at small angles but less so for larger positive and negative angles. Prediction of the turbulence intensity is however relatively poor, which can be attributed to this variable having a lesser impact on the airfoil pressure distribution. Moreover, this variable is not directly set in the CFD simulations as it is used to calculate turbulent boundary conditions (kinetic energy $k$ and specific rate of dissipation $\omega$ of the $k - \omega$ turbulence model), which makes it more difficult to retrieve in this inverse context.

## 6 DISCUSSION

Our results indicate that the type of GNN architecture chosen in the Grouped Reversible Processor has a clear impact on the flow reconstruction quality. From our comparison, we find that, overall, using Graph Attention Network layers usually yields the best reconstructed solutions. However, we also observe that the Graph Isomorphic Network layer is better able to capture detached flows (see Appendix E). Perhaps a combination of the two could lead to better reconstructed flows, which could be for instance implemented by alternating GIN and GAT layers. Lastly, we find that the performance of the GEN layer to be somewhat underwhelming.

Something to note is that the first few layers of nodes surrounding the airfoil carry a disproportionate amount of relevant information which needs to be propagated outwards to an increasing number of nodes, thus creating artificial tree-like paths within the graph. While the Grouped Reversible framework provides a good way to train deep networks in order to circumvent this, other methods may also feasible. One possible solution might be to use hierarchies such as those implemented in Martinkus et al. (2021). Another option could be to apply a message-passing GNN in an iterative

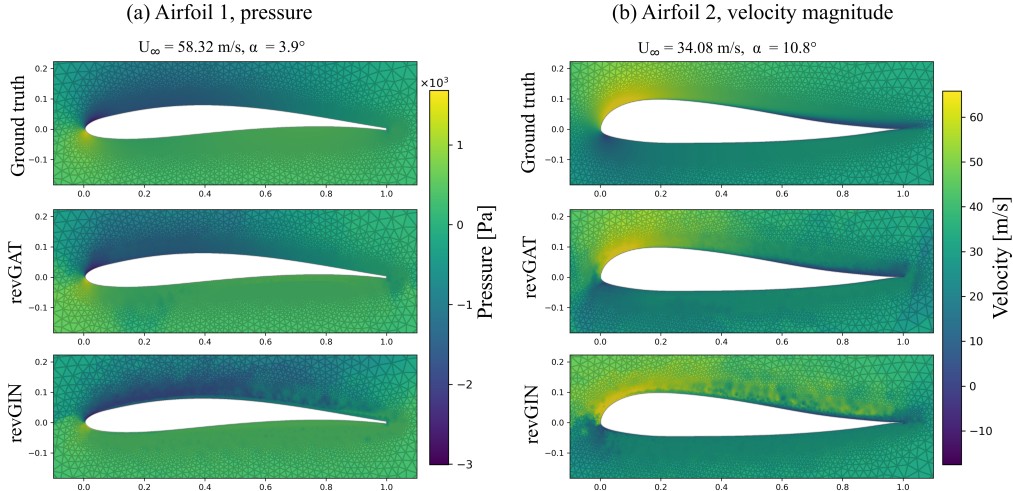

Figure 4: Reconstructed flow around two unseen arbitrary airfoils geometries at different inflow configurations for the revGAT and revGIN models. Figure (a) plots the reconstructed pressure field while the reconstructed velocity field is shown in Figure (b). For each case the ground truth is shown for comparison. Qualitatively, the revGAT model displays fewer artefacts in the solution.

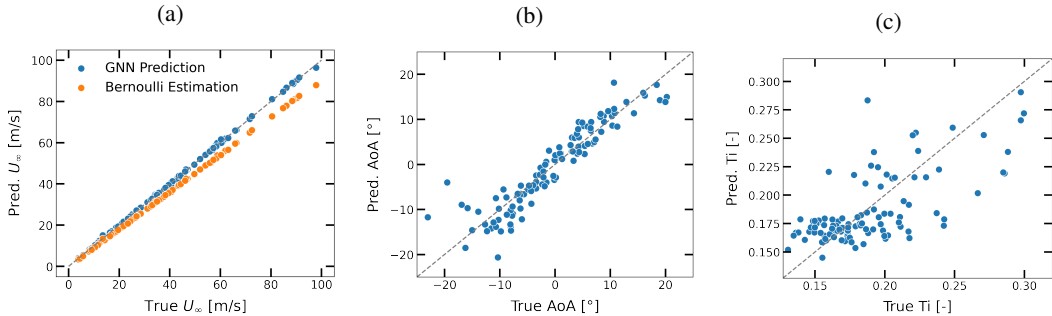

Figure 5: Comparison of the global properties predicted by our revGIN model against the ground truth values, evaluated on the test dataset. The model is able to accurately predict farfield flow velocity thanks to a decent initial estimation (a) and to a lesser extent the angle of attack (b), but it falls short for farfield turbulence intensity prediction (c).

manner, with each step reconstructing increasingly large concentric bands around the airfoil. Physics-driven learning methods could also lead to potential improvements. For instance, the message-passing framework may well be suited to incorporate elements from the Lattice-Bolzmann method (Chen & Doolen, 1998) as it also functions in a similar two step algorithm (collision and streaming). Another possible option would be to minimize the gradient of the pressure, as described in Taha & Gonzalez (2021).

## 7 CONCLUSION

In this work we applied deep graph-based learning techniques to reconstruct pressure and velocity fields around arbitrary airfoil geometries subject to high-Reynolds turbulent flows. We show that, despite the challenges posed by this problem, such as the large graphs and the very localized input information, our Flow Reconstruction GNN framework is able to provide good reconstructed solutions, and infer contextual farfield flow information. We compared several message-passing architectures within the Grouped Reversible Processor GNN, and found that Graph Attention Network layers yielded the best reconstructed solutions. This work provides a flexible framework which may easily be applied to other mesh-based inverse physics problems, and which may be of significant interest to a number of engineering applications.

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

APPENDIX

## A  BENCHMARKING OF THE CFD MODEL

To ensure that the simulations which constitute our dataset are of sufficient quality, we benchmark our CFD pipeline against results reported in the literature for the NACA 0012 airfoil (Krist, 1998; Gregory & O'reilly, 1970). In Figure 6 we plot the pressure coefficient for this airfoil at two angles of attack. We see that our CFD pipeline matches well with both the previous experimental and numerical results.

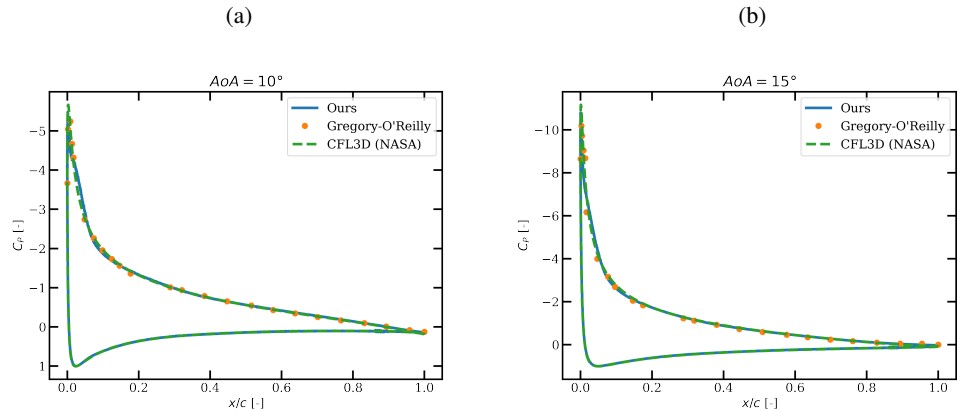

Figure 6: Comparison of our CFD data pipeline against experimental (Gregory & O'reilly, 1970) and other CFD results (Krist, 1998). We plot the pressure coefficient distribution on the surface of a NACA 0012 airfoil at $Re = 6 \cdot 10^6$ at an angle of attack of (a) 10° and (b) 15°.

## B  ADDITIONAL HYPERPARAMETER DETAILS

For all models, 30 layers were used in conjunction with a latent space size of 128. The GAT model is implemented with 2 attention heads. Moreover a LayerNorm layer is applied to the output of each GAT layer, as it was found that this greatly aided training stability. Both the GIN and GEN implementation makes use of ReLu activated MLPs with two hidden layers and LayerNorm. These choice of parameters ensure that each individual layer has approximately 4.5k to 5k learnable parameters. In total for the baseline models with 30 layers and a latent space size of $N = 128$, we obtain models with around 700k trainable parameters.

## C ABLATION OF ESTIMATED GLOBALS

We plot in Figure 7 the result of removing the estimated farfield conditions ($\hat{U}_\infty$ and $\hat{C}_n$) from the input of the model. We note a drastic penalty in the quality of the prediction of farfield quantities.

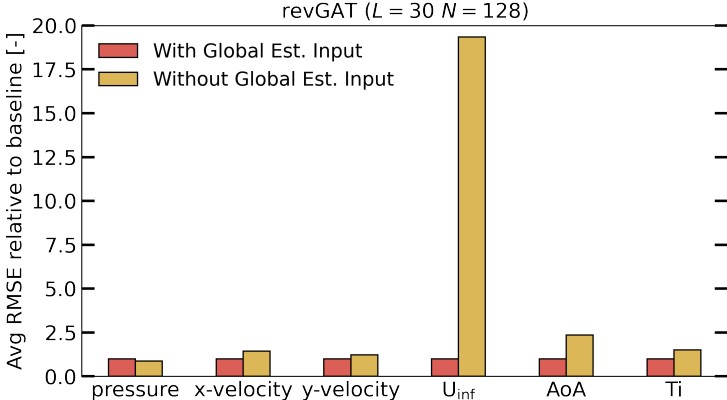

Figure 7: Relative performance of the model when no estimated farfield parameters are used in the input compared to the baseline model.

## D DEPTH AND WIDTH STUDY

We display in Figure 8 the effect of model depth on the reconstruction performance of each reversible model. In Figure 9, we plot the relative performance of different latent space sizes (the width) for the different models.

## E ADDITIONAL RESULTS

We plot in Figure 10 and Figure 11 some additional results, showcasing the reconstructive abilities of our models, as well as some configurations which are not well captured. Notably, detached flows are not always properly reconstructed, an issue which may stem from either the dataset not holding enough detached flows, or from challenges arising from the GNN architecture.

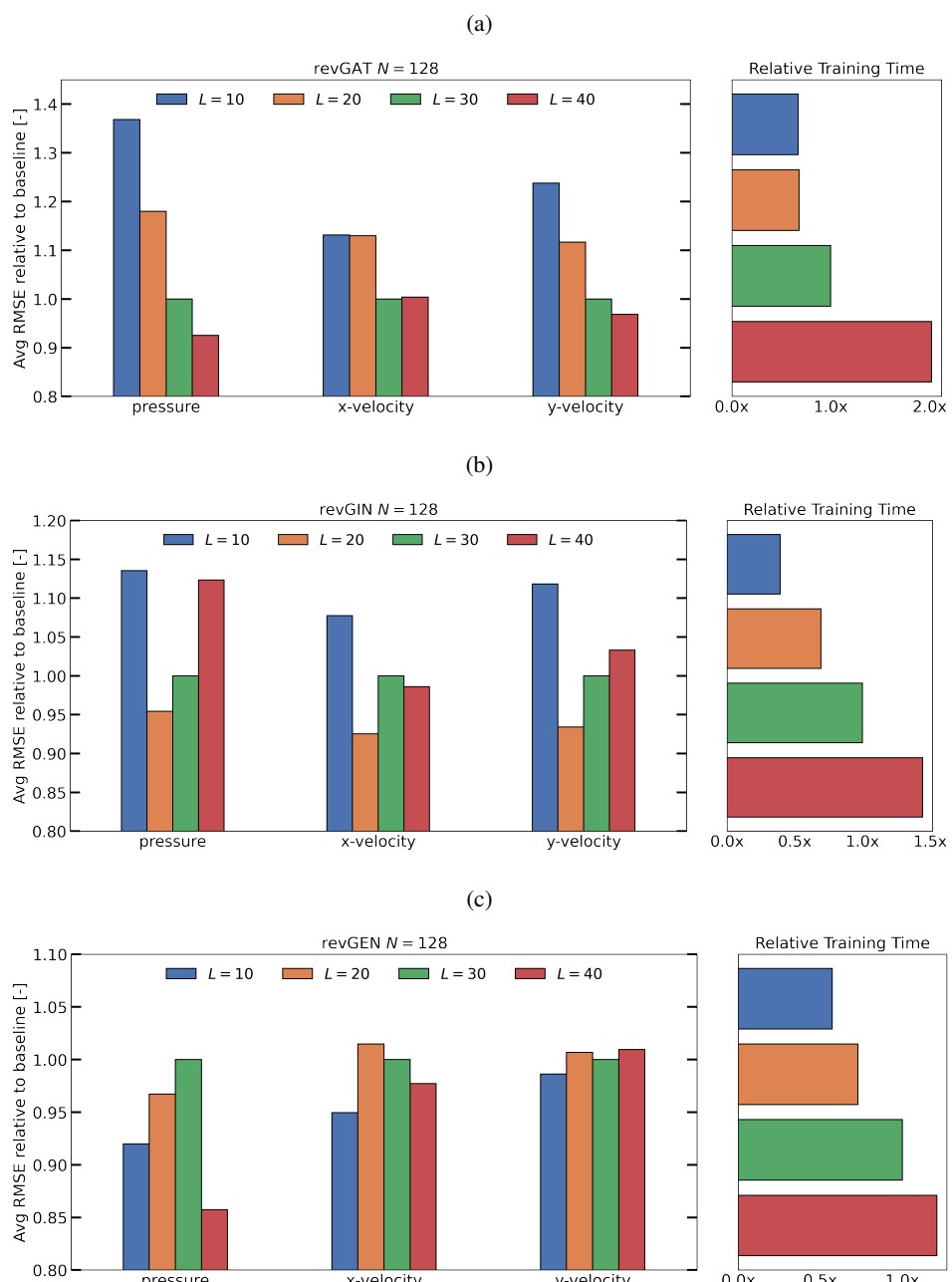

Figure 8: Comparison of the impact of the depth on reconstruction performance for the different layers, relative to the baseline architectures with 30 layers (in green). We also show the relative training time for each depth.

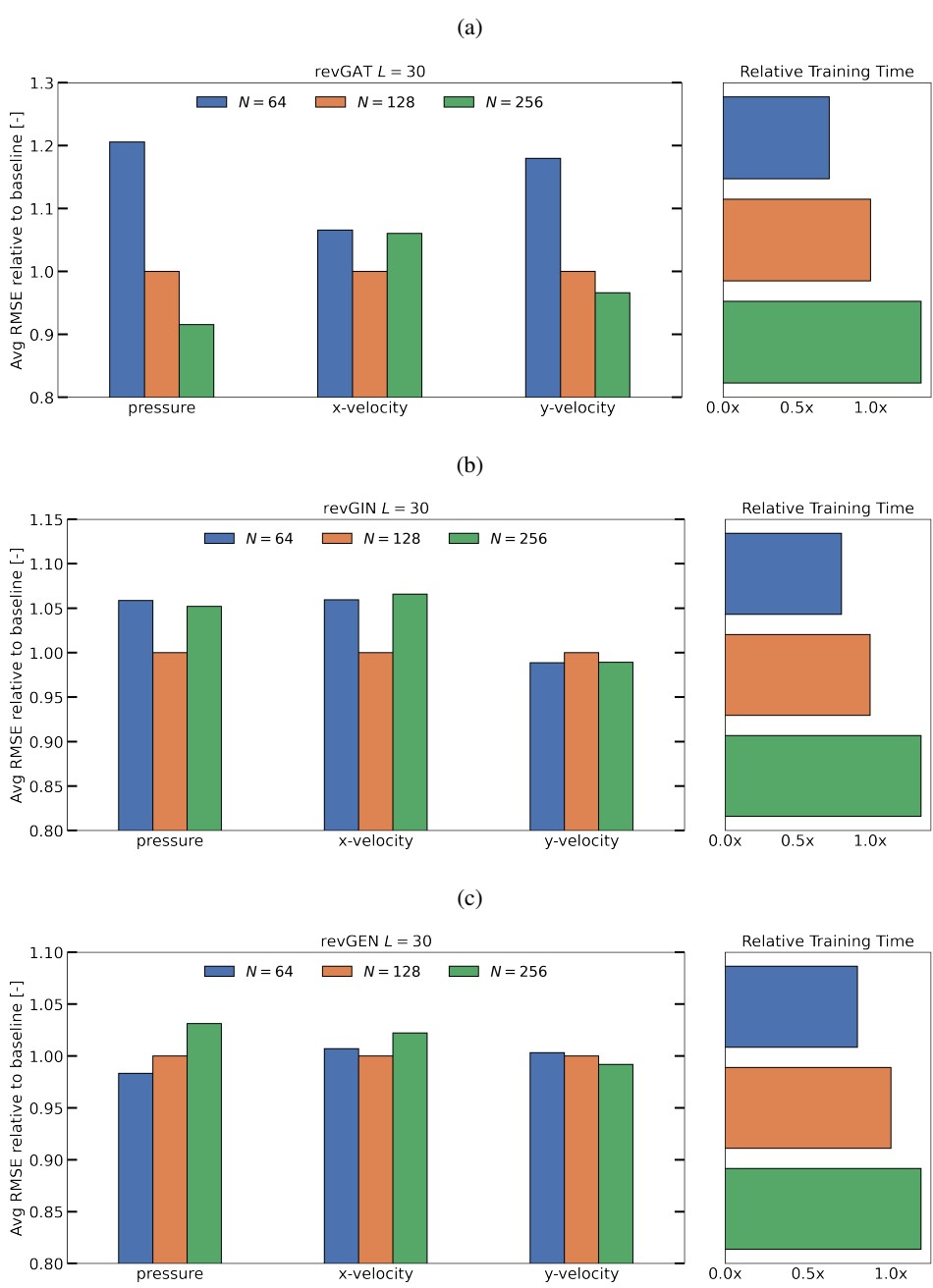

Figure 9: Comparison of the impact of the size of the latent space on reconstruction performance for the different layers, relative to the baseline architectures with a latent space size of $N = 128$ layers (in orange). We also show the relative training time for each width.

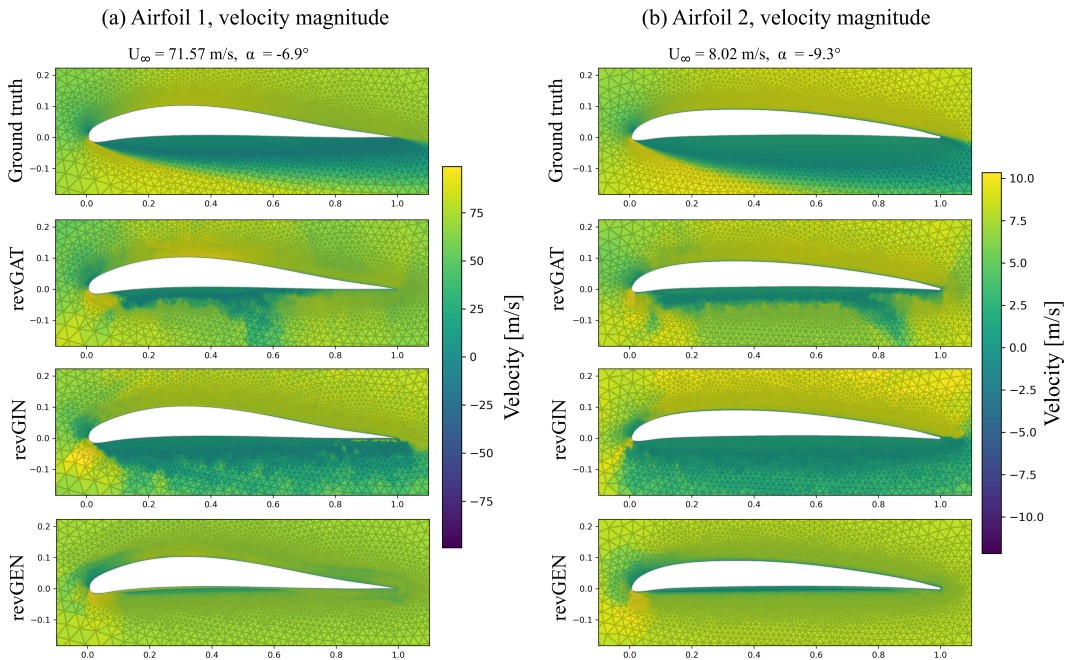

Figure 10: Reconstructed velocity field magnitude around two unseen arbitrary airfoils geometries at different large wake inflow configurations for the revGAT, revGIN and revGEN models. For each case the ground truth is shown for comparison. Qualitatively, the revGIN model is able to better capture the detached flows in the solution.

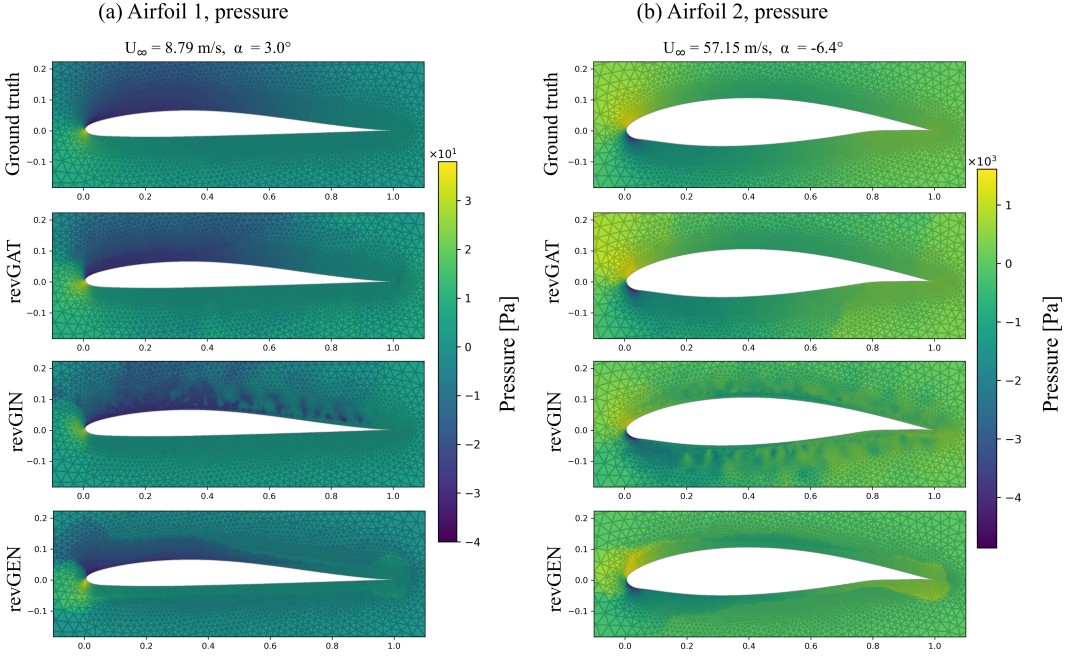

Figure 11: Reconstructed pressure field around two unseen arbitrary airfoils geometries at different inflow configurations for the revGAT, revGIN and revGEN models. For each case the ground truth is shown for comparison. Qualitatively, the revGAT model displays fewer artefacts.

