# OpenReview forum: "Graph Neural Networks for Aerodynamic Flow Reconstruction from Sparse Sensing"
_ICLR.cc/2023/Conference — Submitted to ICLR 2023_

### Official Review · Reviewer_TYBR · 2022-10-20

**Confidence:** 4
**Correctness:** 3
**Technical Novelty And Significance:** 2
**Empirical Novelty And Significance:** 2
**Recommendation:** 3

**Clarity, Quality, Novelty And Reproducibility:**

The paper is well written and clear. Novelty is limited for both the dataset (many existing works released what I think are similar datasets), and the model (the proposed approach seems to be the concatenation of two existing methods).
Reproducibility seems possible, yet code release would be very much appreciated.

**Strength And Weaknesses:**

The paper focuses on a difficult but very interesting problem, with direct applications in aviation. Reconstruction of flows from sparse measurements has already been studied in Erichson et al. but in a very simple scenario. The task introduced by the authors is much more challenging, and the proposed solution seems reasonable. Following the structure of the paper, I organized my questions and remarks in two sections dealing with the dataset first, then the model. These are sorted in ascending order of importance.

About the dataset

- I would have appreciated more statistics about the dataset, including its total weight, average number of nodes and edges, etc.

- It seems that the simulations were carried out under the assumption of incompressibility. It seems to me that this assumption does not hold for flow on an airfoil (Sanchez-Gonzales et al. 2020). Can the authors discuss the relevance of this assumption?

- However, I am quite skeptical about the novelty of the dataset. The authors simulate the evolution of the velocity and pressure field over a large number of different wing profiles and under a large set of initial conditions. However,  this type of dataset already exists in other works. In particular, [1] seems to me to be the closest, but other variants exist [2], [3] (non-exhaustive list). Would it be possible to complete the related works section of the article by comparing the new dataset with the existing works?

About the model

- In appendix B, the models are limited to 4.5k parameters. It seems very low. In comparison, MeshGraphNet (Sanchez-Gonzalez et al. 2020) contains 1.5M parameters for L=15 layers of GNN (but the model is designed for dynamic forecasting). Is this a typo? Are GNN weights shared across layers?

- Section 4.2 presents an auxiliary loss enforcing the divergence of the velocity field to be close to zero. This is perfectly justified in the case of an incompressible simulation. Nevertheless, it is not always respected by the simulator. Have the authors verified that the null divergence hypothesis is verified on their dataset? Also, what is the effect of ablating this term on model performances?

- In general, I deplore the absence of more baselines to compare with the proposed model, which novelty is rather limited. I wonder in particular what is the effect of the grouped reversible GNN relatively to a more traditional solution such as chaining layers of GAT/GIN/GEN?

- The model seems to use a simulation bias to obtain a first approximation of the magnitude of the velocity farfield (the fact that the simulation uses a null pressure field at the boundary). So I have three questions: (1) To what extent could this approximation be performed on real data? (2) How far is this first approximation from the true velocity farfield (this could be checked by plotting figure 5a for this approximation)? and (3) what is the contribution of this first estimation in the performance of the models? in other words, does the performance deteriorate noticeably without this approximation?

I nevertheless point out the relevance of the experiments carried out, in particular table 2 and figure 5 which correctly analyze the important points of the model. I noted several citation errors in the references, and 11 pre-prints among the 33 citations.

[1] Thuerey, N., Weißenow, K., Prantl, L., & Hu, X. (2020). Deep learning methods for Reynolds-averaged Navier–Stokes simulations of airfoil flows. AIAA Journal, 58(1), 25-36.

[2] Ahmed, S., Kamal, K., Ratlamwala, T. A. H., Mathavan, S., Hussain, G., Alkahtani, M., & Alsultan, M. B. M. (2022). Aerodynamic Analyzes of Airfoils Using Machine Learning as an Alternative to RANS Simulation. Applied Sciences, 12(10), 5194.

[3] Schillaci, A., Quadrio, M., Pipolo, C., Restelli, M., & Boracchi, G. (2021, January). Inferring functional properties from fluid dynamics features. In 2020 25th International Conference on Pattern Recognition (ICPR) (pp. 4091-4098). IEEE.

**Summary Of The Paper:**

The paper introduces a new dataset simulating the steady airflow on an airfoil. The dataset contains a large number of different geometries and initial conditions. The authors are then interested in estimating the velocity field and the global context of the simulation from the sole knowledge of sparse measurements of the pressure field on the surface of the airfoil. The proposed method rely on feature propagation and grouped reversible GNN.

**Summary Of The Review:**

Despite the good quality of the work presented, my first impressions lead me to believe that the novelty is limited. Beyond that, some design choices seem questionable to me, and I hope the authors can provide us with convincing justifications. In that case, I may increase my grade.

---

> ### Author Response · Authors · 2022-11-19
> **Response to Reviewer TYBR Part 1**
>
> We thank the reviewer for their constructive comments and questions. We have responded to each remark below.
>
> >  I would have appreciated more statistics about the dataset, including its total weight, average number of nodes and edges, etc.
>
>
> We have added some more information regarding the statistics of the dataset in the final paragraph of the data generation pipeline.
>
> >  It seems that the simulations were carried out under the assumption of incompressibility. It seems to me that this assumption does not hold for flow on an airfoil (Sanchez-Gonzales et al. 2020). Can the authors discuss the relevance of this assumption?
>
> In this work, we only used flows with a maximum velocity of 100m/s. For air, this corresponds to a Mach number of around 0.3, which is the upper limit for a flow to still be considered as incompressible.
>
> >  However, I am quite skeptical about the novelty of the dataset. The authors simulate the evolution of the velocity and pressure field over a large number of different wing profiles and under a large set of initial conditions. However, this type of dataset already exists in other works. In particular, [1] seems to me to be the closest, but other variants exist [2], [3] (non-exhaustive list). Would it be possible to complete the related works section of the article by comparing the new dataset with the existing works?
>
>
>  We have examined the datasets that you have suggested. [1] is indeed the closest to our work, where Gmsh and OpenFOAM are used in conjunction with random boundary condition selection, however our dataset differs in two ways. First, we make the data directly available in graph format instead of an interpolated grid which significantly reduces spatial resolution close to the airfoil. This graph-parsing step requires a decent amount of processing to obtain from the raw OpenFOAM simulations. Secondly, we use a k-omega SST RANS model, which better captures the laminar-to-turbulent transition location of the boundary layer instead of the Spalart-Allmaras model. [2] only considers airfoils of the NACA family at a fixed Reynolds number and angle of attack and [3] only considers 4 NACA airfoil shapes. We have added this information in the related work section.
>
> >  In appendix B, the models are limited to 4.5k parameters. It seems very low. In comparison, MeshGraphNet (Sanchez-Gonzalez et al. 2020) contains 1.5M parameters for L=15 layers of GNN (but the model is designed for dynamic forecasting). Is this a typo? Are GNN weights shared across layers?
>
> For each individual message-passing layer, we set around 4.5k trainable parameters. For our baseline with L=30 layers, with the encoding and decoding layers, this corresponds to a total of around 700k trainable parameters. While this is relatively low compared to the MeshGraphNet model, we also have an order of magnitude more nodes than they typically have, thus we have had to reduce the number of parameters to make the model trainable on a single GPU.
>
> >  Section 4.2 presents an auxiliary loss enforcing the divergence of the velocity field to be close to zero. This is perfectly justified in the case of an incompressible simulation. Nevertheless, it is not always respected by the simulator. Have the authors verified that the null divergence hypothesis is verified on their dataset? Also, what is the effect of ablating this term on model performances?
>
> In our training data, we only use simulations where the continuity is sufficiently converged. Typically, we obtain solutions where the continuity residual is well below 1e-8 and global mass flow errors are usually on the order of 1e-6.
> The divergence-free loss component was only tested. Unfortunately it resulted in better farfield prediction metrics, but tended to increase the amount of artefacts in the reconstructed flows. We have changed the paragraph regarding the loss to mention this and have otherwise removed all reference to this loss component.
>
> >  In general, I deplore the absence of more baselines to compare with the proposed model, which novelty is rather limited. I wonder in particular what is the effect of the grouped reversible GNN relatively to a more traditional solution such as chaining layers of GAT/GIN/GEN?
>
> While we did try other models, such as the Equivariant GNN, results were poor. This is due to the fact that we were limited by GPU memory requirements and therefore could not train GNNs deep enough to propagate information from the airfoil boundary to the rest of the graph while still retaining a large latent space. Only with the Grouped Reversible model were we able to reach 30 message-passing steps with a sufficiently large latent space.

---

> > ### Author Response · Authors · 2022-11-19
> > **Response to Reviewer TYBR Part 2**
> >
> >
> > >  The model seems to use a simulation bias to obtain a first approximation of the magnitude of the velocity farfield (the fact that the simulation uses a null pressure field at the boundary). So I have three questions: (1) To what extent could this approximation be performed on real data? (2) How far is this first approximation from the true velocity farfield (this could be checked by plotting figure 5a for this approximation)? and (3) what is the contribution of this first estimation in the performance of the models? in other words, does the performance deteriorate noticeably without this approximation?
> >
> >  (1) The null pressure field at the boundary is set to avoid having to account for the reference atmospheric pressure offset, which is common practice in CFD. In a real setting, if one is using absolute barometers, a calibration period before deploying the sensors would be necessary. This should however not affect the initial estimation of the farfield velocity significantly, especially when the inflow velocity is high. (2) The initial estimation is consistently off by a few m/s, especially at higher velocities. We have added these estimations to Fig 5a. (3) We also have added a small ablation study in the Appendix which demonstrates the degradation in prediction quality when no estimated farfield conditions are used in the input.
> >
> > > The paper is well written and clear. Novelty is limited for both the dataset (many existing works released what I think are similar datasets), and the model (the proposed approach seems to be the concatenation of two existing methods). Reproducibility seems possible, yet code release would be very much appreciated.
> >
> > Thank you for your review and the pertinent remarks. We believe that there is novelty in the overall method, and that the way in which we combine the different existing elements may be very useful to researchers working on inverse physics ML problems. GNN methods which learn from and replace forward-physics simulators have been a trending topic, but there is relatively little work on inverse problems such as flow reconstruction. We have also attached the code and data necessary to train our model.

---

> > > ### Comment · Reviewer_TYBR · 2022-11-21
> > > **Answer**
> > >
> > > I thank the authors for their responses.
> > >
> > > Ultimately, this paper is hard to evaluate. On one hand, the authors' remarks allow me to see the improvements of the new dataset compared to the existing one. I take full measure of the difficulty of accurately simulating a dataset of this size.
> > > On the other hand, novelty remains limited. it is a more accurate version of already existing datasets. It is not clear how this dataset is more relevant for estimating the performance of a model on an inverse problem, compared to [1]. I am not sure that working with more precise simulations is decisive in these models.
> > >
> > > I have the greatest difficulty in evaluating the model. It seems that the proposed architectures rely heavily on existing results, and the contributions of the authors on this aspect are difficult to identify, apart from the application to a flow reconstruction task. The authors' response suggests that one of the major interests lies in the ability of these models to handle meshes with a very large number of points, which is not feasible with GNN-based baselines. However, this is not highlighted in the paper.
> > >
> > > So far, I'm still leaning towards rejection, but now I have doubt about my understanding of the paper. May I ask the authors to list one last time very explicitly:
> > >  - What is/are the contributions of this new dataset compared to SOTA, not only in terms of dataset quality, but also with regards to the advantages of such quality for the task of interest,
> > >  - What are the contributions made to the architecture presented in the paper and how are they relevant for the task. I am particularly interested by the novelties you added to the existing grouped reversible GNN, and what makes them useful to solve the task.
> > >
> > > Thank you.

---

> > > > ### Author Response · Authors · 2022-11-25
> > > > **Reply**
> > > >
> > > > Thank you for your response and your questions.
> > > >
> > > > > What is/are the contributions of this new dataset compared to SOTA, not only in terms of dataset quality, but also with regards to the advantages of such quality for the task of interest.
> > > >
> > > > This dataset is different to other existing propositions most notably by the fact that we make the simulations available in a graph format. By doing so, we do not interpolate and degrade the resolution of the simulations close to the airfoil. This is necessary for the flow reconstruction problem, as a low resolution on the airfoil surface will lead to poor extrapolation. Moreover, by doing so, we preserve accurate useful secondary information related to engineering quantities (lift and drag coefficient). Another difference with other available datasets is the choice of RANS model.
> > > >
> > > >
> > > > > What are the contributions made to the architecture presented in the paper and how are they relevant for the task. I am particularly interested by the novelties you added to the existing grouped reversible GNN, and what makes them useful to solve the task.
> > > >
> > > > We would like to stress that the novelty lies in the way in which we tackle the flow reconstruction problem using existing GNN methods. The first major hurdle is the lack of node features in the initial graph where we only have known inputs for at best only 5% of the graph nodes (the airfoil nodes). Instead of just initializing the 95% of unknown nodes with random values, we used Feature Propagation to obtain decent initial solutions for the pressure field. The next big obstacle lies in the localization of the known information: it is all contained at the center of the graph on the airfoil nodes. This entails that a very deep network is required to propagate the known information as far out as possible from the airfoil surface, as each layer only transmits information in a 1 hop neighborhood. However training a deep GNN for large graphs is difficult: one usually has to trade off width for depth. The Grouped Reversible framework gives us a way to tackle this issue. While we did not modify the Grouped Reversible GNN per se, we did however test it with the GINE layer, something which we did not see in previous literature.
> > > > Another novel aspect of our work is the prediction of farfield boundary conditions through the use of graph level attributes, as in other mesh-based physics GNN methods this is given as an input.
> > > > Finally, we would like to insist on the fact that no other publication has tried to solve the flow reconstruction problem for arbitrary geometries in arbitrary flow conditions using limited and sparse pressure data on the surface of the airfoil.

---

### Official Review · Reviewer_up38 · 2022-10-24

**Confidence:** 4
**Correctness:** 3
**Technical Novelty And Significance:** 3
**Empirical Novelty And Significance:** 1
**Recommendation:** 3

**Clarity, Quality, Novelty And Reproducibility:**

The paper is easy to read. The novelty is limited to application of ML for CFD analysis. The experimental evaluation is a bit shallow and there are not enough details available to determine reproducibility of the work. The code showing the mesh construction and openFOAM configuration would be helpful.

**Strength And Weaknesses:**

Strengths

- A rich dataset is generated with graph representations of fluid around the airfoil taking 60,000 nodes, different inflow conditions (flow velocity, turbulence, and angle of attack) and different airfoil shapes from the UUIC database.

- The study not only explores different shapes but does a decent sweep of inflow conditions ranging across a large interval of Reynolds number (2 * 10^5 to 6.5 * 10^6). Non-laminar flow (beyond 5 * 10^5) makes flow reconstruction difficult.

- The idea of using global context features in addition to node features is interesting. Incorporating the length of the boundary between cells as an edge feature is also neat. But could authors discuss why the analysis was limited to 1 chord circle centered on the manifold? Could the scalability challenge be elaborated with some quantitative analysis?

Weaknesses

- In absence of access to code, it is not feasible to determine if the meshing by Gmsh and using OpenFOAM has yielded trustworthy simulation data. There is some mention of details such as sizing field but the reviewer is unsure about fidelity of the dataset. Typically, tools such as Ansys Fluent and Comsol are used because of their ability to provide guidance in running CFD on different shapes. Even then, human expertise is critical in CFD analysis. Since the paper mentions dataset as a critical component, the absence of code and dataset for review makes it difficult to determine the quality and hence significant of this part of the paper.

- The experimental results appear to be mixed. The proposed approach fails on farfield turbulence intensity prediction and struggles with prediction of AoA.  It appears the training set was 896 large and the testing was done on 112 airfoils. It is important to make sure that the training and test set had airfoils that were sufficiently different from each other. It will be useful to describe how this split was done. Considering many different splits will also be important. Using a parametric airfoil shape generator can enable scaling the experiments further. Also, is this method restricted to 2D? Why not test the method also on 3D examples?



**Summary Of The Paper:**

The paper considers the problem of extrapolating and reconstructing the flow around 2-D airfoils using measurements on the surface. A deep reversible graph neural network is trained to perform this reconstruction

**Summary Of The Review:**

Overall, the paper takes a standard GNN approach and uses that for CFD analysis. Some of the doubts of the reviewer are attempting to ascertain the value of this work in the CFD domain, since the novelty from the point of view of ML is limited to application. The paper appears to be a work in progress.

---

> ### Author Response · Authors · 2022-11-19
> **Response to Reviewer up38**
>
> We thank the reviewer for their constructive comments and questions. Each comment has been addressed below.
>
> > The idea of using global context features in addition to node features is interesting. Incorporating the length of the boundary between cells as an edge feature is also neat. But could authors discuss why the analysis was limited to 1 chord circle centered on the manifold? Could the scalability challenge be elaborated with some quantitative analysis?
>
> The decision to limit the reconstruction to a 1 chord circle was made due to technical limitations (mainly GPU memory), but mainly because most of the interesting flow physics happens close to the airfoil, as the full CFD domain has a 100 chord radius to allow for sufficient wake dissipation. By just using a GPU with more memory, it should be quite easy to reconstruct larger domains (node features would be increasingly similar further from the airfoil), but not very interesting to do so.
>
> >  In absence of access to code, it is not feasible to determine if the meshing by Gmsh and using OpenFOAM has yielded trustworthy simulation data. There is some mention of details such as sizing field but the reviewer is unsure about fidelity of the dataset. Typically, tools such as Ansys Fluent and Comsol are used because of their ability to provide guidance in running CFD on different shapes. Even then, human expertise is critical in CFD analysis. Since the paper mentions dataset as a critical component, the absence of code and dataset for review makes it difficult to determine the quality and hence significant of this part of the paper.
>
> To answer this remark, we have added to the Appendix a section in which we compare the pressure distribution of a NACA 0012 airfoil simulated with our pipeline against experimental results and other established CFD results.
>
> >  The experimental results appear to be mixed. The proposed approach fails on farfield turbulence intensity prediction and struggles with prediction of AoA. It appears the training set was 896 large and the testing was done on 112 airfoils. It is important to make sure that the training and test set had airfoils that were sufficiently different from each other. It will be useful to describe how this split was done. Considering many different splits will also be important. Using a parametric airfoil shape generator can enable scaling the experiments further.
>
> This is a good point. The current dataset split is random, but it is extremely unlikely that two airfoils with the same geometry end up being simulated with the same boundary conditions, which yields completely different node attributes. While we did consider using a parametric airfoil generator (NACA, PARSEC, etc.), we opted for the UIUC database to get some of the more unique shapes which are usually not parameterizable, such as the very large airfoils and very thin airfoils.
>
> >  Also, is this method restricted to 2D? Why not test the method also on 3D examples?
>
>  At the moment the method is only trained on 2D airfoils. While this is definitely a shortfall for real applications, we justify this by the fact that it is not uncommon to work with 2D flows in aerodynamic engineering applications (e.g. wind turbine blade design). In a real setting, it is also difficult to instrument a large section of a wing/blade with pressure sensors, but a small quasi-2D strip is achievable.

---

> > ### Comment · Reviewer_up38 · 2022-11-30
> > **Thank you for the response.**
> >
> > > The decision to limit the reconstruction to a 1 chord circle was made due to technical limitations (mainly GPU memory), but mainly because most of the interesting flow physics happens close to the airfoil,  .... not very interesting to do so.
> >
> > The reviewer is not in complete agreement with this observation. For simple shapes, this might be true, but the entire exercise of using ML and the purpose of this dataset appears to explore new shapes. The reviewer does not know of a technical argument to be convinced that 1 chord circle is sufficiently good limit.
> >
> > > To answer this remark, we have added to the Appendix a section in which we compare the pressure distribution of a NACA 0012 airfoil simulated with our pipeline against experimental results and other established CFD results.
> >
> > The paper mentions dataset as a critical component, the absence of code and dataset for review makes it difficult to determine the quality and hence significant of this part of the paper. The appendix addition is useful, but just one simulation does not answer the question that whether the meshing was good enough to yield trustworthy simulation data.  The reviewer has experience using OpenFOAM, Ansys Fluent and Comsol, and finds the absence of discussions of challenges of using OpenFOAM with Gmsh meshing a bit perplexing.
> >
> > > "The experimental results appear to be mixed...." This is a good point. The current dataset split is random, .... we opted for the UIUC database to get some of the more unique shapes which are usually not parameterizable, such as the very large airfoils and very thin airfoils.
> >
> > To clarify, the expectation of the reviewer is not that the network has completely memorized specific training data, but rather that the experimental setup proposed by the reviewer would better evaluate generalization. The reviewer is not sure which data points in UIUC dataset are not very closely approximated using parameterized representations. But this is a minor point (because this concern can be raised for many papers).
> >
> > > At the moment the method is only trained on 2D airfoils. While this is definitely a shortfall for real applications ... In a real setting, it is also difficult to instrument a large section of a wing/blade with pressure sensors, but a small quasi-2D strip is achievable.
> >
> > In reviewer's experience, manual design makes heavy use of CFD followed by wind tunnel experiments with scaled 3D models. At least for rotorcraft blades, it is very difficult to ignore 3D. But different design teams and organizations follow different approaches. Putting in some citation (e.g. technical reports from industry) for the claims about the realism of the 2D version of the problem would be useful.

---

### Official Review · Reviewer_N3A3 · 2022-10-24

**Confidence:** 4
**Correctness:** 3
**Technical Novelty And Significance:** 2
**Empirical Novelty And Significance:** 2
**Recommendation:** 6

**Clarity, Quality, Novelty And Reproducibility:**

The paper is clear, consistently written. The degree of novelty is a little vague. Good information is provided for reproducibility.

**Strength And Weaknesses:**

Strengths: Original problem, and a solution that provides quite good inference.

Weaknesses:  The update scheme in the graph could be seen in terms of the Lattice Boltzmann models, except the flow update is defined in an a way that does not automatically lead to zero divergence in summing up the velocities to each vertex to zero. Perhaps relaxation has to be done for a faster convergence. What happens, if the solution is just propagated with from the airfoil surface to the outer edge with the Boltzmann lattice update rules? Also, I think more work should be done on calculating the training data with a LES solver that goes to sub graph network "mesh" sizes, so that the complexity of the features could be used to find a closure for the turbulence and give more reliable solutions for the turbulence, as this is the fundamental problem at high Reynolds numbers.

**Summary Of The Paper:**

The manuscript is describing how to estimate the flow of air around an airfoil only from the pressure and velocity distribution on the airfoil for high Reynolds number flows. Graph based structure is used to define a "mesh" has some intelligence, i.e. features at each vertex. The paper combines far-field information with contextual qualities of the incoming field. A digital ensemble of training data is created with OpenFoam and RANS turbulence models.

The far-field conditions are initialised with Bernoulli flow and the pressure with a random sample with a mean and standard deviation of the values on the airflow. The pressure values are propagated with about 20 feature propagation steps. Different architectures are studied.

The trained Graph model do give quite good velocity profiles, although the turbulence and flow detachment point seem to diverge. From global features, especially the far field velocity, is estimated very well.

.

**Summary Of The Review:**

The paper reproduces the state of the art of standard CFD codes, with the addition that the inflow velocity is kept open. The benefits of this solution should be brought more clearly forward - like how much faster this compared to a normal CFD analysis of the problem - especially as the turbulence prediction seems to be off.

---

> ### Author Response · Authors · 2022-11-19
> **Response to Reviewer N3A3**
>
> Thank you for your review and your remarks.
>
> It would indeed be interesting to use LES data to train the model but this would require a much finer mesh and therefore a much denser graph, which we cannot currently do due to technical limitations (mainly GPU memory).
>
> While the primary goal was not to replace CFD, the GNN method is indeed much much faster than a typical CFD simulation, something which we did not highlight.

---

### Official Review · Reviewer_FKyW · 2022-10-24

**Confidence:** 4
**Correctness:** 4
**Technical Novelty And Significance:** 2
**Empirical Novelty And Significance:** 2
**Recommendation:** 3

**Clarity, Quality, Novelty And Reproducibility:**

The paper is organized well and easy to understand. The authors combine existing ML techniques to solve the proposed problem, but the paper itself does not present any general ML innovations. The used setup is more general than prior work in that the model is trained in conditions of high Re, and with different airfoil shapes at varying angles of attack. The released code and data should make it possible to reproduce the results.

Questions/suggestions:
- Isn't the absolute value of measured pressure potentially important to identify the physical situation one is in? You state that you normalize pressure features to avoid "data leakage". Could you please explain a bit more what you meant here?
- In addition to the absolute error values in Tab. 1 and 2, it would be interesting to see what are the relative errors that the model makes, and whether they vary with parameters such as the Reynolds number (it's conceivable that more turbulent flows are harder to correctly predict).
- In all figures plotting the pressure/velocity fields generated by the models and the ground truth values, it would be helpful to also plot the difference between the ground truth and the model predictions.


**Strength And Weaknesses:**

Strengths of the paper include the use of varying experimental conditions (airfoil shapes, angles of attack, Re) and high Reynolds numbers. The paper also contributes the dataset of simulations used to train the model, and the code used for the experiments is available.

A limitation of the experiments is that only 2d, steady state cases were evaluated. Real settings of interest where pressure sensors are mounted on a physical airfoil would generally be 3d and time-dependent. The experimental exploration done is also limited to testing the 3 layers -- there is no report of testing of the different hyperparameters of the model (L, N, C). The physics-based divergence-free loss component is mentioned, but then not discussed further, so it's unclear what role it plays.


**Summary Of The Paper:**

The paper proposes to use MPNNs to reconstruct the (time-averaged) pressure and velocity fields around an airfoil based on measurements of the pressure distribution at the airfoil's surface. Some global parameters of the flow (farfield velocity, angle of attack, turbulence intensity) are also predicted. The authors use Grouped Reversible GNNs with three different layer types and with Feature Propagation. The models are trained in a supervised manner using a set of over 1k FVM simulations, and in evaluations, the GIN and GAT layers perform better than the GEN layer.


**Summary Of The Review:**

This is an interesting application paper. By tackling a wider range of conditions in a single model, this paper moves towards more realistic scenarios than prior work. However, with all simulations restricted to 2d steady-state conditions, this still feels quite far from the practical applications with MEMS sensors mentioned in the abstract as motivation. With no ML innovations, and lack of clarity whether the relative performance of the different GNN layers in this context has any wider applications, I think that this paper could be better suited to a more specialized venue.

---

> ### Author Response · Authors · 2022-11-19
> **Response to Reviewer FKyW**
>
> We thank the reviewer for their constructive comments and questions about our work. We have tried to address each concern below.
> >  A limitation of the experiments is that only 2d, steady state cases  were evaluated. Real settings of interest where pressure sensors are  mounted on a physical airfoil would generally be 3d and time-dependent.
>
> While it is definitely true that some real applications would benefit from 3D capabilities, many  aerodynamic engineering applications (e.g. wind turbine blade design) tend to consider 2D airfoils with some 3D corrections, which is why we limit our analysis to 2D. The proposed method is only trained on averaged flows, yet these could be seen as snapshots from a time-varying solution. It should be relatively easy to expand the method that we propose here to transient cases, under the condition that there is no heaving of the airfoil (this would require incorporating a history component to account for the aerodynamic hysteresis).
>
> > The experimental exploration done is also limited to testing the 3 layers -- there is no report of testing of the different hyperparameters of the model (L, N, C)
>
>   We have added in the Appendix a study of the depth vs. width for each reversible model.
>
> > The physics-based divergence-free loss component is mentioned, but then not discussed further, so it's unclear what role it plays.
>
> Thank you for picking this up, the divergence-free loss component was indeed tested. It resulted in better farfield prediction metrics, but tended to increase the amount of artefacts in the predicted reconstructed flows. We have changed the paragraph regarding the loss to state this and have otherwise removed all other mentions of this loss component.
>
> > Isn't the absolute value of measured pressure potentially important to identify the physical situation one is in?
>
> You are correct, the absolute offset value of the pressure will play a role, albeit of decreasing importance as the farfield velocity increases. In CFD it is common practice to set the absolute offset to zero. For real applications, if possible some calibration should be done to deal with this.
>
> > You state that you normalize pressure features to avoid "data leakage". Could you please explain a bit more what you meant here?
>
> We normalize the full target pressure field using the maximum of the input pressure measured on the airfoil surface. We do this to avoid having to compute training dataset statistics, which would then also be used during inference. Using the term ‘data leakage’ was incorrect, as in some sense we are actually trying to avoid biasing the model by using input-only normalization. We have replaced this sentence.
>
> > In addition to the absolute error values in Tab. 1 and 2, it would be interesting to see what are the relative errors that the model makes, and whether they vary with parameters such as the Reynolds number (it's conceivable that more turbulent flows are harder to correctly predict).
>
> As our target values can be zero, we do not use relative metrics (numerical values would explode).
>
> > In all figures plotting the pressure/velocity fields generated by the models and the ground truth values, it would be helpful to also plot the difference between the ground truth and the model predictions.
>
> This is also a good remark that we will take into consideration for future work.
>
> > This is an interesting application paper. By tackling a wider range of conditions in a single model, this paper moves towards more realistic scenarios than prior work. However, with all simulations restricted to 2d steady-state conditions, this still feels quite far from the practical applications with MEMS sensors mentioned in the abstract as motivation. With no ML innovations, and lack of clarity whether the relative performance of the different GNN layers in this context has any wider applications, I think that this paper could be better suited to a more specialized venue.
>
> Thank you for your review and the pertinent remarks. We believe that there is novelty in the overall method, and that the way in which we combine the different existing elements may be very useful to researchers working on inverse physics ML problems. GNN methods which learn from and replace forward-physics simulators have been a trending topic, but there is relatively little work on inverse problems such as flow reconstruction.

---

> > ### Comment · Reviewer_FKyW · 2022-11-29
> > **Response to authors**
> >
> > Thank you for your responses and for uploading an updated version of your paper!
> >
> > I have read all the reviews and the responses. This is certainly an interesting paper and general research direction. However, my concern remains that the new 2d simulation dataset coupled with the use of existing techniques for increasing network depth and better initialization, but otherwise without any methodological innovations or generalizable observations, fall a bit short for an ML conference like ICLR.

---

### Decision · Program_Chairs · 2023-01-20

**Decision:**

Reject

**Justification For Why Not Higher Score:**

This is a clear case

**Justification For Why Not Lower Score:**

This is a clear case

**Metareview: Summary, Strengths And Weaknesses:**

This paper on estimating fluid dynamics has received four expert reviews, three of which were highly critical. The main issues raised were a lack of innovation on the ML side (even agreed on by the one very slightly favorable reviewer), lack of reproducability, missing comparisons, and some technical issues were raised wrt to the models. While the authors could address some technical points, the issues on novelty and evaluation remained.

The AC assesses, that the paper is not yet ready for publication and recommends rejection.


**Summary Of Ac-Reviewer Meeting:**

There was no meeting